# Dolutegravir Plus 3TC in Virologically Suppressed PLWHIV: Immunological Outcomes in a Multicenter Retrospective Cohort in Spain during the COVID-19 Pandemic

**DOI:** 10.3390/v15020322

**Published:** 2023-01-24

**Authors:** Luis Buzón, Carlos Dueñas, Roberto Pedrero, Jose Antonio Iribarren, Ignacio de los Santos, Alberto Díaz de Santiago, Miguel Ángel Morán, Guillermo Pousada, Estela Moreno, Eva Ferreira, Alicia Iglesias, Cristina Martín, Julia Gómez, Laura Rodríguez, Miguel Egido, María-Antonia Sepulveda, Jesús Troya

**Affiliations:** 1Infectious Diseases Division, Hospital Universitario de Burgos, 09006 Burgos, Spain; 2Infectious Diseases Division, Hospital Universitario Clínico de Valladolid, 47003 Valladolid, Spain; jduenas@saludcastillayleon.es (C.D.); lrodriguezf@saludcastillayleon.es (L.R.); 3Infanta Leonor University Hospital Research and Innovation Foundation, 28031 Madrid, Spain; robertopedrerotome@gmail.com; 4Infectious Diseases Department, Hospital Universitario de Donostia, 20014 San Sebastián, Spain; antonio.iribarrenloyarte@osakidetza.eus; 5Infectious Diseases Division, Hospital Universitario de La Princesa, 28006 Madrid, Spain; isantosg@hotmail.com; 6Infectious Diseases Division, Hospital Universitario Puerta de Hierro, 28222 Madrid, Spain; ddsalbertorubio@hotmail.com; 7Infectious Diseases Division, Hospital Universitario de Txagorritxu, 01009 Vitoria, Spain; miguelangelmoranrodriguez@gmail.com; 8Infectious Diseases Division, Hospital Álvaro Cunqueiro, 36312 Vigo, Spain; guille.pousada@gmail.com; 9Infectious Diseases Division, Complejo Hospitalario de Navarra, 31008 Pamplona, Spain; estela.moreno.garcia@navarra.es; 10Infectious Diseases Division, Hospital de Segovia, 47002 Segovia, Spain; eferreira@saludcastillayleon.es; 11Infectious Diseases Division, Hospital Universitario de Salamanca, 37007 Salamanca, Spain; aiglesiasg@saludcastillayleon.es; 12Infectious Diseases Division, Complejo Hospitalario de Zamora, 49022 Zamora, Spain; cmartingom@saludcastillayleon.es; 13Infectious Diseases Division, Hospital Universitario Rio Hortega, 47012 Valladolid, Spain; jgomezb@saludcastillayleon.es; 14Infectious Diseases Division, Hospital de Huesca, 22204 Huesca, Spain; megido@salud.aragon.es; 15Infectious Diseases Division, Hospital Virgen de la Salud, 45004 Toledo, Spain; mariaantonia.sepulveda@gmail.com; 16Internal Medicine Department, Hospital Universitario Infanta Leonor, 28031 Madrid, Spain; jesus.troya@salud.madrid.org

**Keywords:** DTG+3TC, switching, viral suppression, immune recovery, safety

## Abstract

Dolutegravir (DTG) based dual therapies for treating PLWHIV are a standard of care nowadays. Switching to DTG and lamivudine (3TC) safety and efficacy were proven in TANGO randomized clinical trial. This multicenter retrospective study included 1032 HIV virologically suppressed patients switching to DTG+3TC from 13 Spanish hospitals. DTG+3TC provided high rates of undetectable viral load over 96%, corresponding to 96.6% (889/921) at 24 weeks, 97.5% (743/763) at 48 weeks, and 98.3% (417/425) at 96 weeks. No significant differences are evident when comparing the total population according to sex, presence of comorbidity, or presence of AIDS. The analysis for paired data showed an increase in CD4+ cell count. A statistically significant increase in CD4+ lymphocyte count was found in those without comorbidities in the three-time series analyzed [average increase at 24 weeks: 48.7 (SD: 215.3) vs. 25.8 (SD: 215.5), *p*-value = 0.050; a mean increase at 48 weeks: 75.1 (SD: 232.9) vs. 42.3 (SD: 255.6), *p*-value = 0.003; a mean increase at 96 weeks: 120.1 (SD: 205.0) vs. 63.8 (SD:275.3), *p*-value = 0.003]. In conclusion, our cohort demonstrates that DTG+3TC is an effective treatment strategy for virologically-suppressed PLWHIV independent of age, sex, and HIV stage, as well as a safe and durable strategy.

## 1. Introduction

For the last 25 years, antiretroviral therapy (ART) for treating people living with HIV (PLWHIV) has stood on three-drug regimens (3-DR). They, most commonly, were designed as a backbone of two nucleos(t)ide reverse transcriptase inhibitors (NRTIs) with a third agent, either a retroviral protease inhibitor (PI), a non-nucleoside reverse transcriptase inhibitor (NNRTI), or more recently, an integrase strand transfer inhibitor (INSTI) [1]. Nevertheless, although current nucleos(t)ide reverse transcriptase inhibitors associated toxicity is remarkably inferior to that associated with the first drugs used for treating PLWHIV in the initial years of the pandemic, current NRTIs still represent a potential burden in terms of short and long-term side effects [2,3]. During the last decade, pursuing strategies with less toxicity led to exploring the possibility of dual therapy.

An initial breakthrough was made by several trials that combined lamivudine (3TC) with different PI [4]. Dolutegravir is a highly active INST against HIV, which remains to date as one of the cornerstones of current strategies for treating PLWHIV [5]. More recently, SWORD randomized clinical trial demonstrated the efficacy and safety of the combination of DTG plus rilpivirine for treatment-experienced PLWHIV with virological suppression [6]. In parallel, the combination of DTG plus 3TC has been approved for treatment-naïve and treatment-experienced PLWHIV after demonstrating its efficacy and safety in the TANGO and GEMINI randomized clinical trials [7,8].

Although DTG plus 3TC has proved its non-inferiority in terms of virological suppression and its durability as a treatment strategy, data regarding its impact on immune activation and inflammation are lacking. Therefore, this study aimed to collect real-life data in a multicenter cohort of PLWHIV treated with DTG plus 3TC as a switch strategy, not only in terms of virological suppression, safety, and durability, but also in terms of its immunological impact through the evolution of the CD4/CD8 ratio in these patients, a marker considered to be a surrogate marker of immune activation and systemic inflammation [9,10]. Data were collected from 1032 patients from 13 Spanish institutions (Spanish cohort of Dolutegravir and 3TC, SPADE-3).

## 2. Materials and Methods

### 2.1. Population and Study Design

From 1 November 2020 to 1 August 2021, a retrospective multicentre study was conducted at 13 hospitals in Spain. Data from 1032 virologically suppressed HIV patients over 18 years old who switched to DTG plus 3TC were collected. Included variables comprise demographics, HIV infection, comorbidities, efficacy in viral suppression, and immune recovery, safety, and tolerability. Since the experimental design requires the study to compare CD4+ and CD8+ lymphocyte counts in individuals at different stages, these data should be considered matched data. Undetectable HIV viral load was considered when <50 copies/mL.

REDcap was used as an online electronic database [11].

### 2.2. Outcomes

The primary outcome (efficacy analysis) was evaluated by determining the percentage of patients with undetectable viral load at weeks 24, 48, and 96 after switching. Virological failure was defined as the presence of an HIV viral load over 50 copies/mL during follow-up.

Secondary outcomes were: (a) patients’ immune recovery evolution through measuring total plasma CD4+/CD8+ cells count and CD4+/CD8+ ratio; (b) adverse events throughout the whole follow-up period; and (c) investigating reasons that led clinicians to switch to DTG plus 3TC.

### 2.3. Statistical Analysis

All statistical analyses were performed using R software (R Core Team, Vienna, Austria, 2021). Data for continuous quantitative variables are presented as medians and interquartile ranges (IQR). Qualitative variables are presented as percentages. Demographics, comorbidities, HIV infection, and diagnosis were compared by gender, the number of comorbidities, and the presence of AIDS. The contrast of medians between the different groups analyzed was compared with parametric (Student’s bilateral *t*-test) and non-parametric (U-Mann-Whitney) tests based on the result of the Kolmogorov-Smirnov normality test. In contrast, the comparison of proportions was carried out using the Chi-square and Fisher tests. In all cases, statistical significance was defined as *p* < 0.05.

Additionally, differences in CD4+ and CD8+ lymphocyte count and CD4+/CD8+ ratio were calculated between the baseline and the three-time points analyzed (24, 48, and 96 weeks of dual DTG plus 3TC treatment) when data were available. Positive values or values greater than zero indicate an increase in the parameter under study. In this case, the average values of each difference are again contrasted according to the three defined population groups and represented graphically in the form of violin plots. Finally, we assessed whether the difference observed at the intra-group level was significant by applying paired means tests (Student’s *t*-test for paired variables; Wilcoxon signed-rank test).

### 2.4. Ethics

The study protocol was approved by the Ethics Committee of Hospital Universitario de Burgos in February 2021 with code 2454 and, subsequently, by the local committee of each institution involved in the study.

## 3. Results

The cohort evaluated 1032 patients from 13 centers in Spain. Baseline characteristics are summarized in Table 1. Men represented 78.6% of the study population with a median age of 48 years (38–57) compared to 53 (45–58) in women. AIDS was diagnosed in 11.4% of patients. At baseline, 44.5% of patients used FTC/TDF as the backbone, and INSTII was the most frequent third drug in 459 (44.5%) patients. Simplification was the leading reason for switching (56.9%), followed by toxicity (16.3%) and drug interaction (5.9%).

The presence of comorbidity was diagnosed in 617 patients (59.8%), and 192 (18.6%) were with three or more comorbidities. Dyslipidaemia was the most frequent one (20.4%), followed by hypertension (11.5%) and chronic liver disease (10.3%). Patients with associated comorbidities were older than those without, *p* < 0.001. AIDS patients had more comorbidities, including hypertension, diabetes, and chronic liver disease.

In addition, 27.9% (192) of overall patients had been diagnosed with hepatitis B virus infection, but only 10/192 (5.3%) presented a positive HBs antigen during the switch to DTG+3TC. On the other hand, 3/56 patients (5.7%) in the comorbidity group showed positive HBs antigen and were on treatment with entecavir. No issues were found within this aspect.

During follow-up, available data provided high rates of undetectable viral load over 96%, corresponding to 96.6% (889/921) at 24 weeks, 97.5% (743/763) at 48 weeks, and 98.3% (417/425) at 96 weeks. No significant differences are evident when comparing the total population according to sex, presence of comorbidity, or presence of AIDS. Nevertheless, there is a tendency related to more virological failures in the groups without comorbidities or AIDS. In those patients with virological failure, no resistance mutations were found when drug resistance tests were performed.

Immune status was assessed through the median baseline CD4+ and CD8+ lymphocyte count and CD4+/CD8+ ratio (Figure 1). No significant differences were found in the CD4+/CD8+ ratio (0.9) during follow-up from baseline to weeks 24 and 96. No statistically significant differences were observed in the study of paired samples based on gender. On the other side, a statistically significant increase in CD4+ lymphocyte count was found in those without comorbidities in the three-time series analyzed [average increase at 24 weeks: 48.7 (SD: 215.3) vs. 25.8 (SD: 215.5), *p*-value = 0.050; a mean increase at 48 weeks: 75.1 (SD: 232.9) vs. 42.3 (SD: 255.6), *p*-value = 0.003; a mean increase at 96 weeks: 120.1 (SD: 205.0) vs. 63.8 (SD:275.3), *p*-value = 0.003]. When this analysis was performed comparing patients with and without AIDS, a significant reduction (*p* < 0.05) was observed in the number of CD8+ lymphocytes in the subgroup without AIDS at 48 and 96 weeks [−27.8 (SD: 318.3); −54.0 (SD: 266.1)], as well as an increase in CD8+ lymphocyte count in the group with AIDS [91.3 (SD: 358.2); 131.4 (SD: 454.4)].

Regarding immune status, patients with AIDS presented, as expected, a lower lymphocyte count (around 150–200 cells/mm3 less than non-AIDS patients) without paired data analysis. Nevertheless, the absolute value of CD8+ lymphocytes remained constant in both groups. This trend translates into a lower CD4+/CD8+ ratio in the AIDS group up to week 48 of dual DTG plus 3TC therapy.

Adverse events (AEs) were scarce and only reported in 14 patients (1.4%). They included renal toxicity in 5 (0.5%), central nervous system toxicity in 8 (0.8%), and gastrointestinal issues in 1 (0.1%). Only three (0.29%) patients discontinued treatment because of AEs.

## 4. Discussion

The efficacy and safety of DTG plus 3TC for treating virologically-suppressed PLWHIV without resistance mutations against DTG and 3TC were settled by the TANGO randomized clinical trial [12]. Similar results are found in our real-life cohort of more than 1000 patients with a suppression efficacy of over 96% and a rate of adverse events lower than 1.5%. However, in the TANGO trial, DTG plus 3TC branch (n = 369) included only 7% of women, and patients older than 50 accounted for 21%. On the contrary, in our cohort (n = 1032), women represented 21.4% of included patients, and individuals older than 50 accounted for 48.8% of the cohort. This fact provides additional helpful information regarding the safety and efficacy of DTG plus 3TC in this population, which are underrepresented in clinical trials.

Beyond this demographic data, in the intention-to-treat exposed population of the TANGO study, 344 patients (93.2%) had an HIV plasma viral load lower than 50 copies/mL at 48 weeks. In our cohort, in the intention-to-treat, 97.5% of patients achieved viral suppression of fewer than 50 copies/mL. Thus, our study confirms and reaffirms the results depicted in the pivotal clinical trial of DTG plus 3TC as a switch study, lining up with some other real-life data from smaller cohorts with similar results [13,14].

Another relevant issue on behalf of DTG plus 3TC treatment is its safety, as severe adverse effects are rarely reported, leading to durability and infrequent switch to other treatment strategies due to adverse effects. Instead, central nervous system events are the most frequently reported AE leading to treatment discontinuation. This data correlates with the TANGO trial [8] and other real-life studies [15,16,17].

A systematic review of the efficacy and safety of DTG plus 3TC from clinical trials and real-life cohorts [18] revealed that neuropsychiatric symptoms, including anxiety, depression, and insomnia, led to the discontinuation of DTG plus 3TC in 1 to 3% of patients [18,19,20]. In our cohort, only three patients (0.29%) had treatment discontinuation due to severe AEs, and only one (0.09%) was related to neuropsychiatric symptoms.

On behalf of CD4+ T cell count and CD4+/CD8+ ratio after switching, our study shows how CD4+ T cell count increased sequentially and significantly from baseline to 96 weeks. Remarkably, this increase was independent of sex, comorbidities, or pre-existing AIDS infection stage. In addition, a reduction in the absolute CD8+ value is also observed in the non-AIDS group.

Our study was designed and carried out during the most challenging days of the coronavirus disease 2019 (COVID-19) pandemic, requiring extra effort by bedside clinicians. Thus, a simple and user-friendly database was developed with a small number of variables to recruit centers. Unfortunately, it led to several limitations that may have clinical implications. They include the study’s retrospective nature; a lack of clinically interesting data, such as body weight gain; and data regarding archived M184V, commonly missing in medical records. Nevertheless, it has undoubted strengths that complement the clinical trials’ data. First, it represents real-life data in the most challenging time of last decades, the COVID-19 pandemic, in which proper adherence to the treatment was probably tackled by many circumstances, adding additional stress to this treatment strategy in terms of performance. Furthermore, this issue probably makes this cohort a completely differential one, as the circumstances in which this dual strategy has been evaluated, with patients having problems accessing their medication or physicians, were unprecedented. These facts have been well described previously [20], underlining how COVID-19 pushed many healthcare systems to their limits in many unpredictable ways, negatively impacting many aspects of patient’s life, including, of course, PLWHIV [21,22]. Thus, the impressive performance of DTG+3TC in very harsh times adds invaluable information to more regular, structured, and controlled data provided by randomized clinical trials and real-life cohorts. Second, it includes a remarkably higher percentage of women and patients over 50 years old than those included in randomized clinical trials, thus providing additional and necessary data in this commonly underrepresented PLWHIV. Third, these cohort results strengthen the results of the TANGO trial, and last but not least, it suggests a potential immunological benefit in increasing CD4+ lymphocyte count and CD4+/CD8+ ratio independently of age, sex, HIV stage, and pre-existing AIDS. Thus, it places DTG plus 3TC as a durable and safe treatment strategy, lined up with previously reported data.

## 5. Conclusions

In conclusion, our cohort confirms previous data from the TANGO study. Furthermore, it demonstrates that DTG+3TC is an effective treatment strategy for virologically-suppressed PLWHIV independent of age, sex, and HIV stage, as well as a safe and durable strategy, even under the stress of the COVID-19 pandemic. In addition, a potential immunological benefit in terms of CD4+ lymphocyte count and increased CD4+/CD8+ ratio is also suggested.

## Figures and Tables

**Figure 1 viruses-15-00322-f001:**
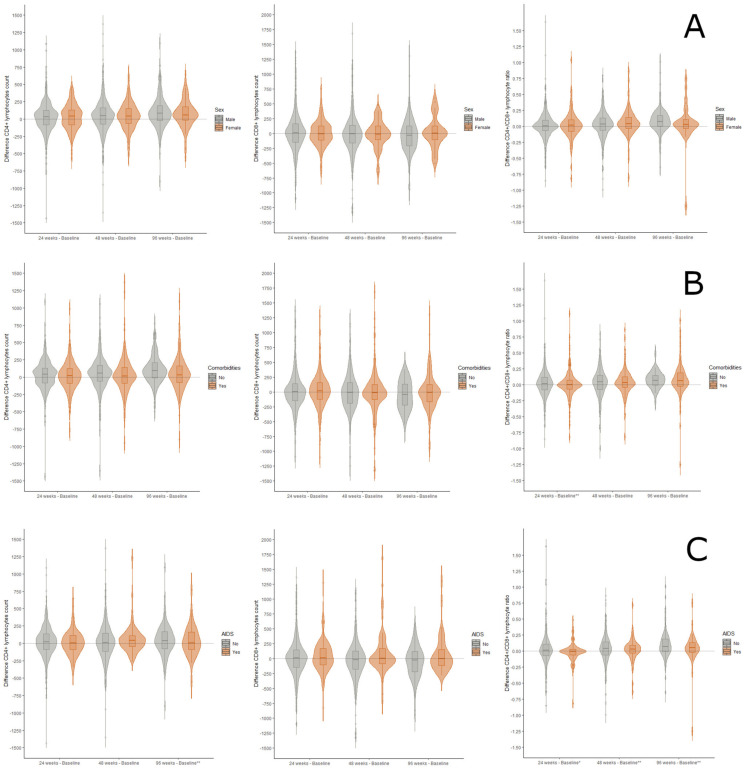
Difference between baseline absolute CD4+ lymphocytes (cells/mm^3^), baseline absolute CD8+ lymphocytes (cells/mm^3^), and baseline CD4/CD8 ratio of HIV-positive patients at 24, 48, and 96 weeks of DTG plus 3TC treatment. Panel (**A**): Gender differences; Panel (**B**): Differences between patients with and without comorbidities; Panel (**C**): Differences in patients with and without AIDS. Significant values: *p*-value < 0.05 = *; *p*-value < 0.01 = **.

**Table 1 viruses-15-00322-t001:** Main characteristics of HIV patients treated with DTG plus 3TC by sex, presence of comorbidities, and AIDS.

	Overall (*n* = 1032)	Female *(n* = 221)	Male (*n* = 811)	*p*-Value	without Comorbidities (*n* = 617)	with Any Comorbidity (*n* = 415)	*p*-Value	No-AIDS Patient (*n* = 634)	AIDS Patient (*n* = 118)	*p*-Value
**DEMOGRAPHICS**										
**Age, median [IQR]**	50.0 [40.0, 57.0]	53.0 [45.0, 58.0]	48.0 [38.0, 57.0]	<0.001	45.0 [36.0, 54.0]	54.0 [47.0, 59.5]	<0.001	47.0 [37.2, 55.0]	56.0 [53.0, 62.0]	<0.001
**Age of HIV diagnosis, median [IQR]**	37.0 [27.0, 47.0]	38.0 [29.0, 49.0]	36.0 [26.0, 46.0]	0.126	33.0 [24.0, 42.0]	42.0 [30.0, 52.0]	<0.001	34.0 [24.0, 45.0]	45.5 [32.0, 53.8]	<0.001
**Male, n (%)**	811 (78.6)	-	-	<0.001	491 (79.6)	320 (77.1)	0.384	512 (80.8)	89 (75.4)	0.229
**Spanish nationality,** **n (%)**	764 (76.9)	156 (73.2)	608 (77.8)	0.186	460 (77.7)	304 (75.6)	0.492	405 (66.9)	97 (82.2)	0.001
**COMORBIDITIES, *n* (%)**										
**Arterial hypertension**	119 (11.5)	26 (11.8)	93 (11.5)	0.997	-	-	-	83 (13.1)	34 (28.8)	<0.001
**Diabetes**	50 (4.8)	9 (4.1)	41 (5.1)	0.67	-	-	-	35 (5.5)	15 (12.7)	0.007
**Dyslipidaemia**	211 (20.4)	48 (21.7)	163 (20.1)	0.663	-	-	-	168 (26.5)	41 (34.7)	0.085
**Heart Disease**	29 (2.8)	4 (1.8)	25 (3.1)	0.432	-	-	-	19 (3.0)	9 (7.6)	0.030
**Cerebrovascular disease**	9 (0.9)	2 (0.9)	7 (0.9)	1.000	-	-	-	4 (0.6)	5 (4.2)	0.004
**Peripheral vascular disease**	11 (1.1)	2 (0.9)	9 (1.1)	1.000	-	-	-	8 (1.3)	3 (2.5)	0.518
**Kidney failure**	40 (3.9)	6 (2.7)	34 (4.2)	0.417	-	-	-	28 (4.4)	12 (10.2)	0.020
**Osteoporosis/Osteopenia**	31 (3.0)	13 (5.9)	18 (2.2)	0.009	-	-	-	24 (3.8)	7 (5.9)	0.409
**Chronic pulmonary disease**	48 (4.7)	14 (6.3)	34 (4.2)	0.246	-	-	-	36 (5.7)	11 (9.3)	0.196
**Psychiatric disorders**	78 (7.6)	23 (10.4)	55 (6.8)	0.096	-	-	-	60 (9.5)	17 (14.4)	0.144
**Cancer**	14 (1.4)	5 (2.3)	9 (1.1)	0.325	-	-	-	10 (1.6)	4 (3.4)	0.334
**Chronic liver disease**	106 (10.3)	30 (13.6)	76 (9.4)	0.089	-	-	-	71 (11.2)	35 (29.7)	<0.001
**Number of comorbidities, *n* (%)**										
**One**	617 (59.8)	126 (57.0)	491 (60.5)	0.439	-	-	-	315 (49.7)	26 (22.0)	<0.001
**Two**	220 (21.3)	45 (20.4)	175 (21.6)		-	-		181 (28.5)	37 (31.4)	
**Three**	105 (10.2)	24 (10.9)	81 (10.0)		-	-		79 (12.5)	25 (21.2)	
**Four**	54 (5.2)	17 (7.7)	37 (4.6)		-	-		35 (5.5)	18 (15.3)	
**Five**	29 (2.8)	8 (3.6)	21 (2.6)		-	-		19 (3.0)	10 (8.5)	
**Six**	4 (0.4)	0 (0.0)	4 (0.5)		-	-		4 (0.6)	0 (0.0)	
**HIV INFECTION**										
**Transmission pathways, *n* (%)**										
**Sexual intercourse**	684 (67.8)	118 (53.6)	566 (71.7)	<0.001	455 (75.5)	229 (56.4)	<0.001	426 (69.4)	64 (54.7)	<0.001
**Intravenous drug injectors**	191 (18.9)	61 (27.7)	130 (16.5)		81 (13.4)	110 (27.1)		84 (13.7)	35 (29.9)	
**Immune status, median [IQR]**										
**Baseline CD4+ (cells/mm^3^)**	753.0 [549.0, 977.0]	763.0 [590.5, 985.0]	744.0 [543.0, 975.8]	0.358	752.5 [551.5, 980.0]	759.0 [531.0, 975.0]	0.944	786.5 [596.5, 1005.8]	604.0 [404.5, 933.0]	<0.001
**24 weeks CD4+ (cells/mm^3^)**	770.5 [592.8, 980.0]	785.0 [601.0, 986.5]	766.0 [592.0, 970.8]	0.482	766.0 [596.5, 976.5]	773.0 [590.0, 982.0]	0.933	808.0 [630.5, 1000.5]	644.0 [449.0, 875.5]	<0.001
**48 weeks CD4+ (cells/mm^3^)**	782.0 [574.0, 1004.0]	779.0 [606.0, 978.0]	784.5 [567.0, 1012.5]	0.996	789.0 [589.0, 1034.0]	776.0 [559.5, 948.8]	0.179	801.0 [591.5, 1029.5]	628.0 [424.2, 866.0]	<0.001
**96 weeks CD4+ (cells/mm^3^)**	823.0 [613.2, 1048.0]	802.0 [652.0, 1037.0]	839.0 [604.5, 1051.0]	0.854	831.0 [633.5, 1058.5]	803.0 [560.2, 1036.5]	0.306	851.0 [678.0, 1125.0]	649.0 [440.5, 856.2]	<0.001
**Baseline CD8+ (cells/mm3)**	867.5 [630.0, 1179.5]	805.0 [589.5, 1084.5]	878.0 [653.0, 1196.5]	0.048	880.0 [632.0, 1156.0]	861.0 [627.0, 1198.0]	0.752	875.0 [637.5, 1199.5]	827.0 [609.0, 1100.0]	0.141
**24 weeks CD8+ (cells/mm3)**	897.0 [656.0, 1220.0]	792.0 [603.0, 1188.5]	913.0 [674.0, 1247.0]	0.020	895.0 [677.5, 1244.5]	898.0 [636.0, 1211.2]	0.793	899.5 [656.8, 1241.0]	871.0 [635.0, 1134.0]	0.517
**48 weeks CD8+ (cells/mm3)**	908.0 [638.5, 1229.8]	817.0 [533.0, 1119.0]	922.0 [661.5, 1248.5]	0.020	910.0 [639.5, 1257.5]	907.0 [640.0, 1208.0]	0.897	900.0 [635.5, 1220.5]	959.5 [603.2, 1224.8]	0.651
**96 weeks CD8+ (cells/mm3)**	906.0 [628.5, 1241.5]	922.0 [625.0, 1222.5]	906.0 [634.5, 1268.5]	0.903	980.0 [673.2, 1280.0]	893.0 [617.0, 1227.0]	0.491	956.0 [672.0, 1246.0]	862.0 [484.5, 1152.5]	0.094
**Baseline CD4+/CD8+ (cells/mm3)**	0.9 [0.6, 1.2]	0.9 [0.7, 1.4]	0.9 [0.6, 1.2]	0.033	0.9 [0.6, 1.2]	0.9 [0.6, 1.3]	0.601	0.9 [0.7, 1.3]	0.8 [0.5, 1.1]	0.003
**24 weeks CD4+/CD8+ (cells/mm3)**	0.9 [0.6, 1.2]	1.0 [0.7, 1.3]	0.8 [0.6, 1.2]	0.011	0.9 [0.7, 1.2]	0.9 [0.6, 1.2]	0.855	0.9 [0.7, 1.2]	0.7 [0.5, 1.1]	0.001
**48 weeks CD4+/CD8+ (cells/mm3)**	0.9 [0.6, 1.2]	1.0 [0.7, 1.4]	0.9 [0.6, 1.2]	0.034	0.9 [0.6, 1.3]	0.9 [0.6, 1.2]	0.635	0.9 [0.7, 1.3]	0.7 [0.5, 1.0]	<0.001
**96 weeks CD4+/CD8+ (cells/mm3)**	0.9 [0.7, 1.3]	0.9 [0.7, 1.3]	0.9 [0.7, 1.3]	0.770	0.9 [0.7, 1.3]	0.9 [0.7, 1.3]	0.981	0.9 [0.7, 1.4]	0.8 [0.7, 1.2]	0.198
**HIV DIAGNOSIS *n* (%)**										
**Previous treatments, *n* (%)**										
**ABC/3TC**	384 (37.2)	77 (34.8)	307 (37.9)	0.458	156 (25.3)	228 (54.9)	<0.001	319 (50.3)	64 (54.2)	0.495
**FTC/TDF**	459 (44.5)	96 (43.4)	363 (44.8)	0.784	193 (31.3)	266 (64.1)	<0.001	367 (57.9)	90 (76.3)	<0.001
**FTC/TAF**	149 (14.4)	22 (10.0)	127 (15.7)	0.042	70 (11.3)	79 (19.0)	<0.001	131 (20.7)	18 (15.3)	0.220
**PI ^**	271 (26.3)	79 (35.7)	192 (23.7)	<0.001	77 (12.5)	194 (46.7)	<0.001	202 (31.9)	66 (55.9)	<0.001
**INSTI**	475 (46.0)	82 (37.1)	393 (48.5)	0.003	229 (37.1)	246 (59.3)	<0.001	407 (64.2)	67 (56.8)	0.153
**NNRTI**	340 (32.9)	82 (37.1)	258 (31.8)	0.161	113 (18.3)	227 (54.7)	<0.001	270 (42.6)	67 (56.8)	0.006
**Reasons for switching, *n* (%)**										
**Toxicity**	168 (16.3)	45 (20.4)	123 (15.2)	0.080	40 (6.5)	128 (30.8)	<0.001	119 (18.8)	48 (40.7)	<0.001
**Drug interaction**	61 (5.9)	22 (10.0)	39 (4.8)	0.007	15 (2.4)	46 (11.1)	<0.001	48 (7.6)	13 (11.0)	0.282
**Simplification**	587 (56.9)	105 (47.5)	482 (59.4)	0.002	269 (43.6)	318 (76.6)	<0.001	493 (77.8)	93 (78.8)	0.895
**Transition therapy to injectable drugs**	9 (0.9)	1 (0.5)	8 (1.0)	0.727	5 (0.8)	4 (1.0)	1.000	9 (1.4)	0 (0.0)	0.400
**Simplicity**	33 (3.2)	5 (2.3)	28 (3.5)	0.499	16 (2.6)	17 (4.1)	0.244	24 (3.8)	8 (6.8)	0.218
**Cost**	26 (2.5)	5 (2.3)	21 (2.6)	0.974	6 (1.0)	20 (4.8)	<0.001	13 (2.1)	13 (11.0)	<0.001
**COINFECTIONS, *n* (%)**										
**HBV diagnosis**	192 (27.9)	36 (27.1)	156 (28.1)	0.904	56 (18.6)	136 (35.1)	<0.001	139 (24.3)	51 (44.0)	<0.001
**HBsAg positive**	10 (5.3)	1 (2.8)	9 (5.9)	0.738	3 (5.7)	7 (5.1)	1	4 (2.9)	6 (11.8)	0.043
**HCV positive ELISA**	160 (23.0)	41 (30.4)	119 (21.2)	0.032	19 (6.3)	141 (35.7)	<0.001	111 (19.2)	49 (42.2)	<0.001
**HCV positive PCR**	52 (34.7)	16 (42.1)	36 (32.1)	0.359	4 (23.5)	48 (36.1)	0.451	26 (25.5)	26 (54.2)	0.001
**VIRAL LOAD < 50 copies/mL, *n* (%)**										
**Baseline**	943 (96.0)	206 (95.4)	737 (96.2)	0.716	565 (95.4)	378 (96.9)	0.318	576 (95.8)	107 (96.4)	0.991
**24 weeks**	889 (96.6)	196 (97.5)	693 (96.4)	0.574	539 (96.6)	350 (96.7)	1.000	532 (96.7)	100 (95.2)	0.638
**48 weeks**	743 (97.5)	162 (96.4)	581 (97.8)	0.463	467 (98.1)	276 (96.5)	0.256	393 (97.3)	81 (94.2)	0.258
**96 weeks**	417 (98.3)	98 (97.0)	319 (98.8)	0.456	304 (99.0)	113 (96.6)	0.181	126 (97.7)	38 (95.0)	0.735

^ In some patients previous treatments included PI in combination con INSTI.

## Data Availability

All data are kept by the investigators of the SPADE-3 project.

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
