# Peer review of "Dolutegravir Plus 3TC in Virologically Suppressed PLWHIV: Immunological Outcomes in a Multicenter Retrospective Cohort in Spain during the COVID-19 Pandemic"

_viruses, 2023, doi:10.3390/v15020322_

Round 1
Reviewer 1 Report
The authors present a large cohort of people living with HIV whose treatment was simplified to DTG+3TC or DTG/3TC. The great contribution of these data lies in the large number of people over 50 years of age, women, and patients with comorbidities included in the study, which reflects a scenario that is usually less represented in clinical trials.
Several minor comments need to be addressed:
1. The authors use PLWHIV and PLHIV (line 29, 53, 58, 62, 67, ...) in various parts of the document. They could unify the terminology.
2. The percentage of patients with different third agents as previous treatments are, all of them, more than 100%, is this correct? Maybe this is not a data of the baseline time but previous treatments anytime. Please clarify the data.
3. The authors use DTG+3TC or DTG/3TC in a similar way. It is probable that many patients used one or the other depending on the availability in the centers. It would be good to specify if the few cases of loss of virological control are or not associated with the absence of STR. Virological failure criteria should be added in "Methods".
4. Could the author specify in how many of the patients RAMs emerged due to loss of virological control (if drug resistance tests could be done?)
5. Had the patients with positive HBs antigen any related problems?
Author Response
Estimated reviewer,
Thanks for your suggestions. We have tried to improve the manuscript according to them.
1. The authors use PLWHIV and PLHIV (line 29, 53, 58, 62, 67, ...) in various parts of the document. They could unify the terminology.
We agree with the reviewer, and we have unified the terminology
2. The percentage of patients with different third agents as previous treatments are, all of them, more than 100%, is this correct? Maybe this is not a data of the baseline time but previous treatments anytime. Please clarify the data.
We agree with the reviewer and have clarified this aspect in table 1. The percentage is over 100% because some patients were on treatments based in two third agents, including bPI and INSTI (line 128)
3. The authors use DTG+3TC or DTG/3TC in a similar way. It is probable that many patients used one or the other depending on the availability in the centers. It would be good to specify if the few cases of loss of virological control are or not associated with the absence of STR. Virological failure criteria should be added in "Methods".
We agree with the reviewer and have used all the paper DTG plus 3TC. Unfortunately, due to the study's retrospective nature, this information is unavailable in patients with a loss of virological control. Virological failure was defined in the methods section (lines 88-89)
4. Could the author specify in how many of the patients RAMs emerged due to loss of virological control (if drug resistance tests could be done?)
As indicated by the reviewer, we have specified this information in the results section (lines 142-143)
5. Had the patients with positive HBs antigen any related problems?
As indicated by the reviewer, we have specified this information in the results section (lines 136)

Reviewer 2 Report
The authors of “Dolutegravir plus 3TC in virologically suppressed PLWHIV: 2 Immunological outcomes in a multicenter retrospective cohort 3 in Spain during the COVID-19 pandemic” demonstrated DTG+3TC maintained or increased viral suppression (viral load <50) over time after switching from the previous viral suppression treatment. CD4 lymphocyte count increased over time among the patients without comorbidities.
Comments:
1. Add the VL undetectable threshold under outcomes in Materials and Methods section.
2. On lines 94-95, t-test compares the means and U-Mann-Whitney test compares the medians. So update the sentence to reflect that. Is the t-test a two-sided or one-sided test?
3. On lines 104-106 and lines 137-138, it seems that the differences (follow-up – baseline) within each sub-group (sex, comorbidity, and AIDs) were compared using the paired t-test or Wilcoxon signed-rank test. Should a comparison of differences among all patients be done before the sub-group comparisons?
4. Beside Figure 1 that shows the difference of CD4/CD8 at each follow-up visit from CD4/CD8 at the baseline not the paired data as stated in the text, authors may consider generating a figure of graphs that include the observations at all visits (the baseline and follow-up visits). The observations within patients are connected by lines to show the trajectory of CD4/CD8 over time within individual patients.
5. It is not clear the sentence “Regarding immune status, patients with AIDS presented, as expected, a lower lymphocyte count (150-200 cells/mm3) without paired-data analysis.” means. Does this sentence refer to the CD4 count for patients with AIDS is 150-200 cells/mm3 lower than the patients without AIDS?
Author Response
Estimated reviewer,
Thank you for your suggestions. We have tried to improve the manuscript according to them.
Response to reviewer 2:
- Add the VL undetectable threshold under outcomes in Materials and Methods section.
As indicated by the reviewer, we have added the viral load undetectable threshold under outcomes in the Materials and Methods section (lines 82-83)
- On lines 94-95, t-test compares the means and U-Mann-Whitney test compares the medians. So update the sentence to reflect that. Is the t-test a two-sided or one-sided test?
In response to the reviewer, both tests are used to compare means or medians. They have been used according to the distribution of the variables to be compared. The student's t-test is bilateral. It has been indicated in the text in the methods section (line 100)
- On lines 104-106 and lines 137-138, it seems that the differences (follow-up – baseline) within each sub-group (sex, comorbidity, and AIDs) were compared using the paired t-test or Wilcoxon signed-rank test. Should a comparison of differences among all patients be done before the sub-group comparisons?
We think that It is unnecessary to compare differences between all patients prior to subgroup comparisons. However, we have chosen to compare directly between subgroups because it yields much more information and because, as shown in Table 1, the starting situation of the subgroups is different.
- Beside Figure 1 that shows the difference of CD4/CD8 at each follow-up visit from CD4/CD8 at the baseline not the paired data as stated in the text, authors may consider generating a figure of graphs that include the observations at all visits (the baseline and follow-up visits). The observations within patients are connected by lines to show the trajectory of CD4/CD8 over time within individual patients.
In response to the reviewer, the information provided on CD4/CD8 is considered complete. Table 1 shows the baseline values at 24, 48, and 96 weeks according to the established population subgroups. In this case, there are hardly any differences in the median values and interquartile range as a function of time. Figure 1 shows a more detailed calculation of the difference between baseline values and those at different times in a paired manner, where slight variations can be seen. We have included information regarding CD4/CD8 ratio during follow-up at baseline, 24, and 96 weeks (lines 145-146)
- It is not clear the sentence “Regarding immune status, patients with AIDS presented, as expected, a lower lymphocyte count (150-200 cells/mm3) without paired-data analysis.” means. Does this sentence refer to the CD4 count for patients with AIDS is 150-200 cells/mm3 lower than the patients without AIDS?
As indicated by the reviewer, we have rewritten the phrase to clarify the information (lines 157-159)
